# Integrative genomic analysis reveals mechanisms of immune evasion in *P. falciparum* malaria

Mame Massar Dieng[1,7], Aïssatou Diawara[1,7], Vinu Manikandan[1], Hala Tamim El Jarkass[1,6], Samuel Sindié Sermé[2], Salif Sombié[2], Aïssata Barry[2], Sam Aboubacar Coulibaly[2], Amidou Diarra[2], Nizar Drou[3], Marc Arnoux [4], Ayman Yousif[3], Alfred B. Tiono[2], Sodiomon B. Sirima[2,5], Issiaka Soulama [2] & Youssef Idaghdour [1✉]

The mechanisms behind the ability of *Plasmodium falciparum* to evade host immune system are poorly understood and are a major roadblock in achieving malaria elimination. Here, we use integrative genomic profiling and a longitudinal pediatric cohort in Burkina Faso to demonstrate the role of post-transcriptional regulation in host immune response in malaria. We report a strong signature of miRNA expression differentiation associated with *P. falciparum* infection (127 out of 320 miRNAs, B-H FDR 5%) and parasitemia (72 miRNAs, B-H FDR 5%). Integrative miRNA-mRNA analysis implicates several infection-responsive miRNAs (e.g., miR-16-5p, miR-15a-5p and miR-181c-5p) promoting lymphocyte cell death. miRNA *cis*-eQTL analysis using whole-genome sequencing data identified 1,376 genetic variants associated with the expression of 34 miRNAs (B-H FDR 5%). We report a protective effect of rs114136945 minor allele on parasitemia mediated through miR-598-3p expression. These results highlight the impact of post-transcriptional regulation, immune cell death processes and host genetic regulatory control in malaria.

[1] Program in Biology, Division of Science and Mathematics, New York University Abu Dhabi, Abu Dhabi, UAE. [2] Centre National de Recherche et de Formation sur le Paludisme, Ouagadougou, Burkina Faso. [3] Bioinformatics Core, New York University Abu Dhabi, Abu Dhabi, UAE. [4] Core Technology Platforms, New York University Abu Dhabi, Abu Dhabi, UAE. [5] Groupe de Recherche Action en Santé, Ouagadougou, Burkina Faso. [6] Present address: Department of Molecular Genetics, University of Toronto, Toronto, ON, Canada. [7] These authors contributed equally: Mame Massar Dieng, Aïssatou Diawara. ✉email: youssef.idaghdour@nyu.edu

Malaria is a major cause of morbidity and mortality with nearly half of the world's population at risk of infection and 435,000 deaths annually[1]. The largest burden of the disease occurs in Sub-Saharan Africa with significant negative socio-economic consequences on the continent where *Plasmodium falciparum* accounts for >99% of estimated cases[1]. The search for effective and sustainable therapeutic strategies for malaria is hampered by limited understanding of the sources of variation in host immune response to natural *P. falciparum* infection. Several factors such as host and parasite genetics, state of immunity, and environment contribute to the observed variation. This in turn leads to variability in the symptomology and clinical manifestations of the disease, ranging from asymptomatic and symptomatic parasitemia to complicated and cerebral malaria[2,3].

Our understanding of the biological mechanisms underlying variation in host immune response to *P. falciparum* infection and in particular immune evasion is poorly understood[4]. Genome-wide association studies (GWAS) have only mapped a few loci contributing to host resistance to malaria[5–9]. The complexity of genotype–phenotype relationships in malaria and the limited power of GWAS in African populations warranted the use of alternative approaches that capture both genetic and environmental effects in vivo. Gene expression profiling has emerged as a powerful approach to study these mechanisms and overcome these limitations[10–13]. A central mechanism in immune response to infection is post-transcriptional regulation[14] and miRNAs in particular have emerged as key regulators of this response[15–17]. In malaria, a few studies have investigated the expression profiles of host miRNAs using either animal models or in vitro settings[18–21]. Using targeted miRNA profiling of adults experimentally infected with *Plasmodium*, Burel et al.[22] reported evidence of a potential role for miRNAs in inter-individual variability in immune response to *P. falciparum* infection. This study also demonstrated that individuals with higher expression of miR-15a-5p, miR-30c-5p, and miR-30e-5p had an increased frequency of activated and proliferating T cells and could better control *P. falciparum* growth during infection. Importantly, given the specificity of host immune processes and the fact that they vary based on host genotype and the environment[11,23], it follows that studying these effects in vivo in African children before and after natural *Plasmodium* infection would be informative.

To address this, we recruited and sampled a longitudinal pediatric cohort in Burkina Faso and completed global miRNA-Seq, total RNASeq profiling experiments and whole genome sequencing (WGS) at 30× coverage. The cohort was comprehensively assessed for various clinical, molecular, and cellular phenotypes before infection, during asymptomatic parasitemia and symptomatic parasitemia, and after treatment (68 samples in the discovery phase). Our hypothesis is that a robust matched design where children are monitored closely and sampled over time will reveal key differentially expressed miRNAs and early transcriptional response associated with biological pathways and functions implicated in in vivo host response to *P. falciparum* infection. The identified signature of miRNAs expression is striking with over one-third of the expressed miRNAs being significantly differentially expressed (127 miRNAs at Benjamini–Hochberg FDR (B–H FDR) 5%) following *P. falciparum* infection of noninfected children. To follow-up on this finding, in the following year we profiled a replication cohort of 53 infected children and captured a host-specific miRNA signature (72 miRNAs, B–H FDR 5%) of a primary blood-stage malaria clinical trait (parasitemia). In total, 36 miRNAs were associated with both infection (noninfected relative to infected individuals) and parasitemia (within the infected group) in the discovery and replication sets.

Using joint analysis of miRNA and mRNA profiles generated separately but from the same blood samples collected from the same individuals, we identified key core adaptive immune pathways and cellular processes dysregulated during infection. In particular, we highlight infection-responsive miR-16-5p, miR-15a-5p, and miR-181c-5p that target the anti-apoptotic gene BCL2, induce apoptosis of immune cells and restrict the development of T cells[24–26]. Finally, genetic mapping of miRNAs using WGS data identified 1,376 genetic variants associated with the expression levels of 34 miRNAs (B–H FDR 5%). Together, these findings suggest that the mechanisms involved in controlling *P. falciparum* infection are under host genetic control at both the transcriptional and post-transcriptional levels.

## Results

**Study population (discovery and replication sets).** A longitudinal pediatric cohort of 150 healthy children was recruited in Banfora, Burkina Faso. Enrollment took place at the end of the dry season and none of the children were *P. falciparum*-infected at the time of the first sample collection. The children were of age 2–10 years, had no disease symptoms or chronic disease history (including sickle cell disease) based on medical records and thorough physical assessment. In this study, a subset of 72 children 88% of whom have hemoglobin genotype AA were included, all of whom were not *P. falciparum*-infected when initially sampled and were subsequently followed on a weekly basis for up to 6 months. For the discovery set, clinical phenotyping, sampling and genomic profiling were done at four time points: (i) before infection (at enrollment), (ii) during asymptomatic parasitemia, (iii) at the beginning of symptomatic parasitemia and (iv) 3 weeks after treatment (discovery set, Fig. 1a). When asymptomatic parasitemia was detected, a closer follow-up of the children was done to sample the early stage of symptomatic infection and immediately followed by Artemether/lumefantrine treatment, as per Burkina Faso's national malaria treatment guidelines. A replication set of infected children was sampled the following year during the wet season (Fig. 1f, see "Methods" for details).

**Dynamics of miRNAs expression during *P. falciparum* infection.** To better understand the temporal dynamics of miRNAs expression during natural *P. falciparum* infection, we generated 68 whole blood miRNAs expression profiles across matched individuals in the four stages sampled: before infection (BI), asymptomatic parasitemia (AP), symptomatic parasitemia (SP) and after treatment (AT) (Fig. 1a) using the TruSeq Small RNA protocol. Stringent bioinformatic analysis and quality control of the miRNA sequencing data resulted in retaining 320 miRNAs for downstream analysis (see Methods for details). Principal component analysis (PCA) showed a strong correlation structure in the data with the first expression principal component (ePC1) clearly capturing the effect of *P. falciparum* infection (Fig. 1b, c). PCA of pair groups captures the gradual temporal changes of *P. falciparum* infection on the correlation structure of the data (Supplementary Fig. S2a). Noninfected and treated samples are largely indistinguishable, clearly showing that the correlation structure of the miRNA transcriptome three weeks after treatment reverts back to before infection status (Fig. 1b, c). This analysis is in large agreement with miRNA differential expression analysis that confirmed the strong stage-specific signature of miRNA expression (Fig. 1e) with 46 miRNAs (14%) significantly up or downregulated (5% B–H FDR) during asymptomatic parasitemia and 97 miRNAs (30%) during symptomatic parasitemia accounting for sex, age and white blood cell count. Breaking down the analysis of covariance into pairwise comparisons shows the gradual change of miRNAs expression and this

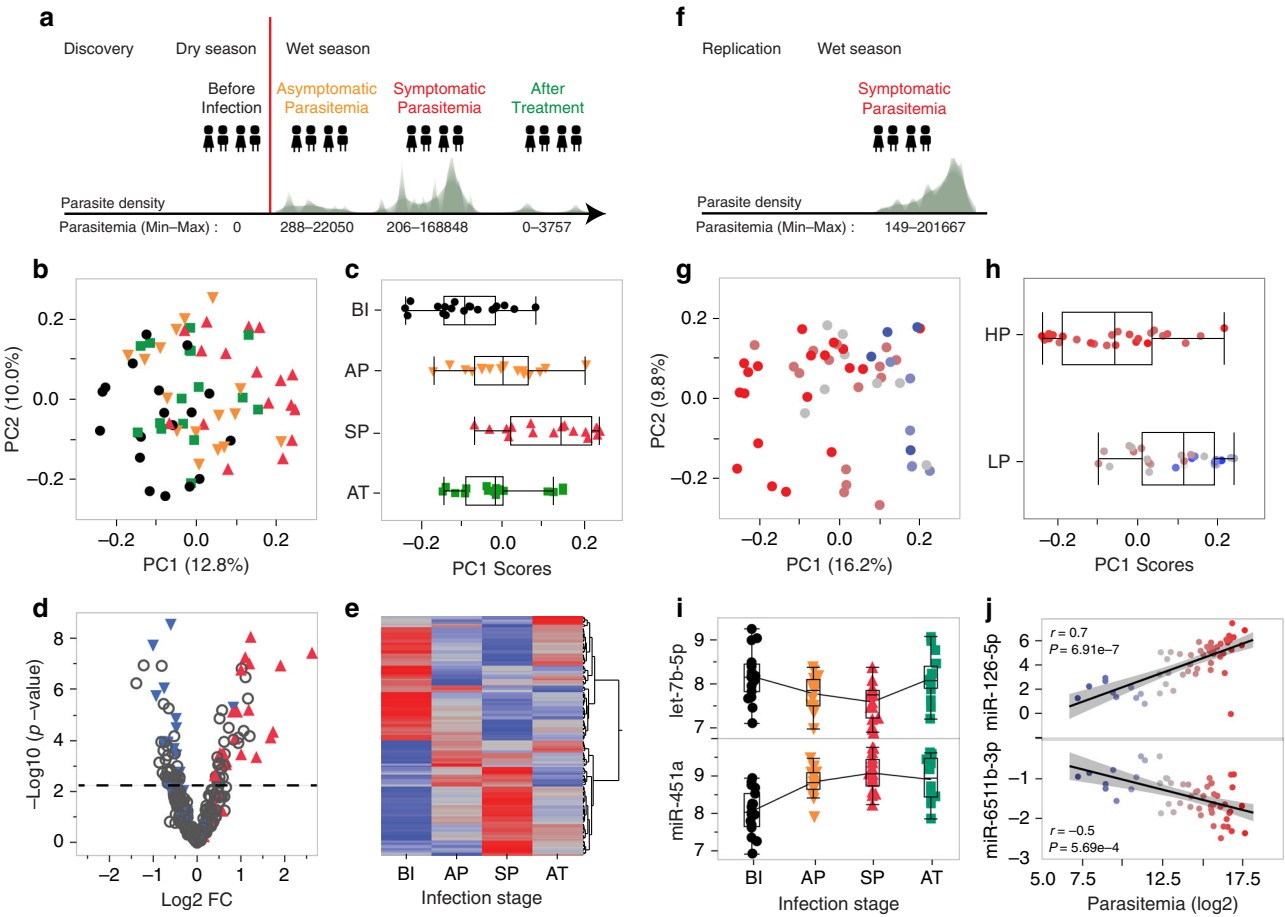

**Fig. 1 Study design and dynamics of global miRNAs expression profiling in malaria. a** Study design and infection stages profiled in the discovery phase: Before Infection (n = 19), Asymptomatic Parasitemia (n = 16), Symptomatic Parasitemia (n = 17), and After Treatment (n = 16). Parasite density plots and the range of parasitemia (minimum and maximum number of parasites per μl of blood) in each stage are shown. **b** PCA of the miRNA transcriptome in the discovery set (n = 68). **c** PC1 scores of the four stages of infection in the discovery set (n = 68): BI (n = 19), AP (n = 16), SP (n = 17), and AT (n = 16). **d** Volcano plot of differential miRNA expression between the SP (n = 17) and BI (n = 19) groups. The magnitude (log2 fold change; x axis) and statistical significance (−log10 P) of differential expression of individual miRNA derived from repeated measures analysis of covariance (see Methods). The horizontal line indicates the −log10 P corresponding to B-H FDR 5%. Seventy-two miRNAs either positively (labeled in red) or negatively (labeled in blue) correlated with parasitemia in the replication set (n = 53, multiple regression) are highlighted. **e** Hierarchical clustering of the standardized least square means of 127 differentially expressed miRNAs across the four stages of infection (repeated measures analysis of covariance, B–H FDR 5%.). **f** Parasite density plot and the range of parasitemia (minimum and maximum number of parasites per μl of blood) in the replication set (Symptomatic Parasitemia; n = 53) are shown. **g** PCA of the miRNA transcriptome in the replication set (n = 53). The color gradient (blue to gray to red) indicates levels of log2 parasitemia (from low to high). **h** PC1 scores for the high parasitemia (HP; n = 26) and low parasitemia (LP; n = 27) groups. High and low parasitemia is determined based on the log2 parasitemia median value of 14.8. **i** Expression values of let-7b-5p and miR-451a across the four stages of infection: BI (n = 19), AP (n = 16), SP (n = 17), and AT (n = 16). **j** Correlation between expression values of miR-126-5p and miR-6511b-3p, and log2 parasitemia (n = 53) adjusting for sex, age and white blood cell count (multiple regression). Pearson correlation coefficient (r) and P values for miR-126-5 (r = 0.7, B–H P = 6.91e-7) and miR-6511b-3p (r = −0.5, B–H P = 5.69e-4) are shown. The 95% confidence interval of the line of best fit is shown in shaded gray. The data presented within each condition (discovery set: BI, AP, SP, and AT; replication set) are from biologically independent samples. In the discovery set, the data is from repeated measurements as shown in Supplementary Data 1. The box plots in **c**, **h**, and **i** show the median, the 25th and 75th percentiles as box edges, and the 5th and 95th percentiles as bounds of whiskers.

trend is visually shown in the volcano plots for each transition in Supplementary Fig. S2b, c and Supplementary Data 3. Contrasting patterns of temporal changes of miRNAs expression levels for miR-451a and miR-let-7b-5p are shown in Fig. 1i. Interestingly, increased expression of miR-451a in erythrocytes has been described as a malaria-protective factor in sickle cell disease[19] and the downregulation of miR-let-7b-5p was reported to activate the innate immune response through the upregulation of TLR4 in response to bacterial infection[27]. Taken together, these results clearly show the strong effect of *P. falciparum* infection on miRNAs expression in a highly coordinated manner.

**Association between miRNAs expression levels and parasitemia in the replication set.** To identify miRNAs associated with infection status in the discovery set and with parasitemia (as one of the main clinical traits of blood-stage malaria), we generated 53 global whole blood miRNA expression profiles from children with symptomatic *P. falciparum* infection presenting a range of parasitemia (Fig. 1f). Stringent alignment and bioinformatic analysis were performed on the data and resulted in the retention of 320 miRNAs for downstream analysis (302 miRNAs overlap between the discovery and replication datasets). PCA of the replication dataset shows a clear trend of the effect of

parasitemia captured in ePC1 explaining 16.2% of the variation in the data (Fig. 1g, h). Multiple regression analysis accounting for age, sex and white blood cell count revealed that 20% (72 out of 320) of expressed miRNAs are associated with parasitemia (B–H FDR 5%, Supplementary Data 3). Figure 1j shows two examples of miRNAs correlated with parasitemia (miR-126-5p, $r = 0.7$, B–H FDR $= 6.91e$-7 and miR-6511b-3p, $r = -0.5$, B–H FDR $= 5.69e$-4). Of the 97 miRNAs differentially expressed during symptomatic parasitemia relative to before infection, 36 miRNAs (38%) are associated with parasitemia. Of these, 35 miRNAs are differentially expressed (up and downregulation) in the same direction as the direction of association (positive and negative) with parasitemia (B–H FDR 5%, highlighted in blue and red in Fig. 1d). These results identified the set of miRNAs that are both responsive to *P. falciparum* infection and associated with parasitemia. We quantified the expression of 16 of these miRNAs using qPCR and validated 12 of them as showing the same trend (two-tailed Student's *t* test, $P < 0.01$), Supplementary Fig. S2d and Supplementary Data 4). The remaining four miRNAs were either not statistically significant or showed the opposite trend. In a stepwise general linear model accounting for sex, age, and white blood cell count and using a twofold cross validation by randomly shuffling the dataset into training and test sets (60 and 40% of the full dataset, respectively), four miRNAs (hsa-miR-199-3p, hsa-miR-3173-5p, hsa-miR-342-3p, and hsa-miR-532-5p) significantly predicted log2 parasitemia in the test dataset ($r = 0.73$, B–H FDR $< 0.05$, RMSE $= 2.1983$, Harrell's C-Statistic $= 0.8238$, see "Methods" and Supplementary Fig. S3).

**Integrative miRNA-mRNA analysis**. To evaluate the biological effects of miRNA expression differentiation during *P. falciparum* infection, we generated global mRNA expression profiles from 51 children during symptomatic parasitemia (see "Methods") and performed integrative analysis of miRNAs and mRNAs profiles as previously done in other studies[28,29]. Given that miRNA-mediated post-transcriptional silencing of targeted mRNA occurs through a combination of mechanisms including mRNA decay, decapping and deadenylation, and translational repression[30], only a subset of mRNA targeted by miRNA are expected to show a correlation between their transcript abundance and that of their corresponding miRNA. Since the action of miRNA results in negative regulation of expression, we focused on identifying anti-correlated miRNA-mRNA pairs using miRNA and mRNA expression data generated from the same set of samples and individuals. This analysis was restricted to experimentally validated and highly predicted miRNA-mRNA pairs (Ingenuity Pathway Analysis, IPA database) and included 133 miRNAs that are differentially expressed after infection with *P. falciparum* and/or associated with parasitemia, and 12,375 mRNA transcripts expressed during symptomatic parasitemia. This analysis revealed 171,737 miRNA-mRNA statistically significant associations (at a relaxed 5% nominal *P*, Pearson correlation) of which 93% are negative correlations (Pearson correlation coefficient ranging between $-0.28$ and $-0.75$ with a median of $-0.32$, Supplementary Data 5). Of these, we identified 1,031 anti-correlated miRNA-mRNA (*P* range 0.04-3.6e-7) predicted (high prediction score) and/or experimentally validated using the miRNA Target Filter tool implemented in IPA. These associations correspond to 456 unique genes targeted by 88 miRNAs. Examples of miRNA-mRNA negative correlations are shown in Fig. 2a (miR-451a-ABCB1, $r = -0.38$, $P = 6.22e$-3; miR-181c-5p-BCL2, $r = -0.44$, $P = 1.33e$-3 and miR-128-3p-LGALS3, $r = -0.57$, $P = 8.21e$-6).

**Functional annotation of miRNA expression differentiation**. We next performed pathway enrichment analysis using IPA to identify the biological functions implicating miRNAs both associated with *P. falciparum* infection and negatively correlated with their corresponding mRNA targets. This analysis implicates miRNA differentiation in the divergence in key cell death and adaptive immune functions. We assessed the molecular functions in play by focusing on the statistically significant enriched functional subcategories (using B–H FDR-adjusted *P* values). The analysis revealed Cell Death and Survival ($P = 1.14e$-3–8.96e-2), Cellular Development ($P = 4.12e$-3–8.89e-2), Cellular Function and Maintenance ($P = 4.12e$-3–7.91e-2), Cellular Growth and Proliferation ($P = 4.12e$-3–8.89e-2), and Cellular Movement ($P = 4.34e$-3–8.91e-2) as the five most significantly enriched (Fig. 2b and Supplementary Data 6). A closer inspection of the molecular and cellular functions of the most significant category, Cell Death and Survival identified Apoptosis ($P = 3.16e$-7), necrosis ($P = 7.84e$-7) and other cell death mechanisms as the most implicated by this analysis (Fig. 2c and Supplementary Data 6). The patterns from multiple pathway and molecular functions enrichment analyses are consistent with each other and indicate that molecular processes of adaptive immune cell proliferation and death are a prime target of differential miRNA regulation during the course of *P. falciparum* infection.

To identify the cascade of upstream miRNA regulators that can explain the observed gene expression changes of the target mRNAs and the significantly enriched functional subcategories identified, we ran upstream regulator analysis using IPA. This analysis identified a list of 39 upstream regulators including miR-16-5p, miR-17-5p, miR-181c-5p, and miR-125b-5p (full list in Supplementary Data 7). Among these, we highlight miR-16-5p given its significant gradual upregulation during the course of *P. falciparum* infection and the role of its target genes in apoptosis, in particular BCL2 which is downregulated during symptomatic parasitemia (Fig. 2d). To identify the cell types most likely subject to apoptosis under the effect of these miRNAs, we compared cell count data obtained at the time of collection. This analysis revealed that lymphocytes are the most depleted cell type during the course of infection both in terms of absolute count and proportion (Fig. 2e, Supplementary Fig. S4a, $P < 0.001$, two-tailed Student's *t* test). Analysis of total RNA data using cell type deconvolution tool Epic[31] confirmed this conclusion and showed that lymphocyte depletion is driven by statistically significant loss of B cells and CD4 T cells among the cell types considered (Supplementary Fig. S4b–h). To test whether miR-16-5p has the ability to modulate cell proliferation, we transfected HEK-293 and Hela cells with mimic-miR-16-5p or a mimic-control. Forty-eight hours after transfection, cell survival/proliferation was assessed using the MTT formazan assay. We observed a statistically significant reduction of HEK and Hela cell proliferation of 17 and 26%, respectively, compared to control-transfected cells ($P < 0.05$, Wilcoxon paired rank test, Fig. 3e). Taken together, these results indicate that miR-16-5p upregulation during *P. falciparum* infection is implicated in the depletion of immune cell populations through apoptosis. In summary, the analyses detailed above provide evidence for a central role of post-transcriptional regulation in early transcriptional response to natural *P. falciparum* infection. A fundamental question that follows pertains to the nature and extent of the contribution of regulatory genetic variation to the observed response. To address this question, we set to examine the association between allelic genetic variation and miRNA transcript abundance.

**Genetic control of miRNAs expression**. To identify genetic variants associated with miRNAs expression levels during *P. falciparum* infection, we mapped local acting *cis*-miR-eQTLs by testing the association between expression levels of miRNAs

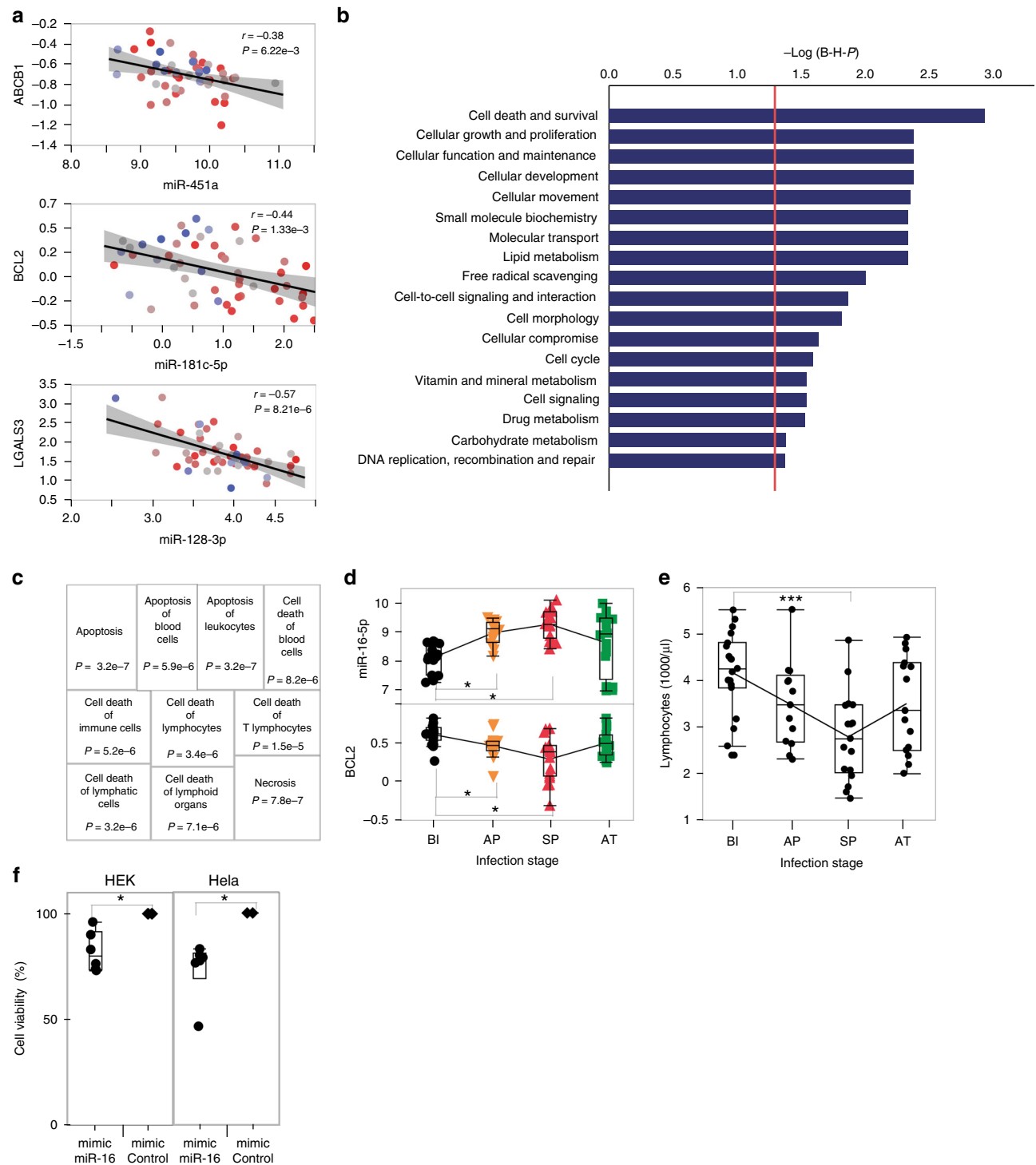

and 129,310 genetic variants from the replication set. The study is underpowered to detect *trans* effect and therefore the analysis was limited to *cis* regulatory effects. The 129,310 genetic variants were obtained from whole-genome sequencing data of the study participants and correspond to variants with MAF > 5% and located within a window of 200 kb centered from the miRNA. On average, each miRNA was tested against 468 SNPs. Given the challenges of genetic mapping in African populations[32] and to further increase statistical power, we used an additive model where two genotypic classes (homozygotes individuals vs individuals carrying at least one alternative allele) are tested against miRNAs expression levels (see Methods for details). In total, we identified

1,376 *cis*-miR-eQTLs for 34 miRNAs (B-H FDR < 0.05, (Fig. 3a, Table 1). The full list of *cis*-miR-eQTLs associations including the results of permutation tests is provided in Supplementary Data 8.

The strongest association (B-H FDR-adjusted $P = 6.73\text{e-}27$) identified was between miR-4326 and SNP rs6062431 (Fig. 3b) located on the precursor RNA at six base pairs from the mature miR-4326, suggesting that allelic variation in miRNA precursor sequence can alter mature miRNA levels. Next, we focused on associations that can pinpoint potential mechanisms through which miRNA response to *P. falciparum* infection can be under regulatory genetic control. For example, children carrying one or two copies of the minor allele rs10741471 have elevated levels of

**Fig. 2 Integrative analysis of miRNA-mRNA and functional annotation. a** Examples of negative correlations between miRNAs and their experimentally validated targets: miR-451a-ABCB1, miR-181c-5p-BCL2, and miR-128-3p-LGALS3 ($n = 51$, multiple regression adjusting for sex, age and white blood cell count). The corresponding Pearson correlation coefficients ($r$) and nominal $P$ are shown. Each individual is colored by its parasite density (from the lowest (blue) to the highest (red). The 95% confidence intervals of the line of best fit are shaded in gray. **b** Molecular and cellular functions obtained from the enrichment analysis of 456 genes targeted by the 88 miRNAs differentially expressed between before infection (BI) symptomatic parasitemia (SP) groups and/or associated with parasitemia. The horizontal bars show the level of significance. $P$ values are calculated and adjusted using a right-tailed Fisher's exact test and Benjamini–Hochberg method. The vertical red line represents the $P$ significance threshold ($-\log (0.05)$). **c** Annotation of subcategories of the most significantly enriched molecular and cellular function: Cell Death and Survival. $P$ values are calculated using a right-tailed Fisher's exact test. **d** Temporal expression changes of miR-16-5p (upper panel) and BCL2 (lower panel) during the course of infection in the discovery set: BI ($n = 19$), AP ($n = 16$), SP ($n = 17$), and AT ($n = 16$). Pairwise comparisons were done using a two-tailed Student's $t$ test (*indicates $P < 0.05$). **e** Temporal changes of absolute lymphocyte counts during the course of infection in the discovery set: BI ($n = 19$), AP ($n = 16$), SP ($n = 17$), and AT ($n = 16$). Pairwise comparisons were done using a two-tailed Student's $t$ test (*$P < 0.001$). **f** Experimental validation of apoptotic role of miR-16-5p using cell proliferation assays of HEK and Hela cells transfected with miR-16-5p mimic ($n = 6$ independent experiments) or a mimic negative control ($n = 6$ independent experiments). Each data point shows cell viability of mimic-miR-16-5p treated cells relative to mimic-miR-control treated cells 48 h after transfection. Statistical significance was assessed using a paired two-tailed Wilcoxon rank test ($P = 0.031$ for both cell lines). The box plots in **d**–**f** show the median, the 25th and 75th percentiles as box edges, and the 5th and 95th percentiles as bounds of whiskers.

miR-1304-3p expression while those homozygous for the major allele have low levels of expression of this miRNA (B-H FDR-adjusted $P = 3.68e-17$). Interestingly, miR-1304-3p has previously been shown to directly target heme oxygenase-1 (HMOX1) encoding the primary heme-degrading enzyme which has been shown to be protective against severe malaria in mice[33]. SNP rs10741471 is in complete LD with a group of SNPs (shown in Fig. 3c) suggesting that the causal genetic effect detected is located within the block and yet to be identified.

An opposite trend is observed with SNP rs114136945 associated with miR-598-3p (B–H FDR-adjusted $P = 7.22e-3$) where the minor allele is associated with lower levels of expression. miR-598-3p is of particular interest as it is positively associated with parasitemia ($r = 0.72$, B–H FDR-adjusted $P = 6.91e-7$) and differentially expressed between the symptomatic parasitemia and before-infection stages (Fold change = 2.26, B–H FDR-adjusted $P = 4.15e-4$). We noted that the SNP itself is significantly associated with parasitemia ($P = 0.002$) (Fig. 3e) warranting a test of mediation for a SNP effect through modulation of miRNAs expression and subsequently regulation of target mRNA. To test for this, we performed mediation analysis for miRNAs correlated with parasitemia (miR-598-3p, miR-126-5p, miR-624-5p, miR-5010-3p and miR-3201) among the significant SNP-miRNA pairs. The analysis tests the association between peak SNPs and parasitemia then performs a test of mediation of miRNA expression using bootstrapping (1,000 simulations) to assess significance (See "Methods"). When significant, the test supports miRNA expression as a mediator of the association between the regulatory genetic variant and parasitemia. This analysis revealed a significant mediation effect of miR-598-3p expression in the association between SNP rs114136945 and parasitemia ($\beta_3 = -3.1$, 95% CI [$-4.5$, $-1.4$] $P = 0.002$, d–f) with a proportion mediated of 84% and strongly supports a protective effect of rs114136945 minor allele through downregulation on miR-598-3p expression.

## Discussion
Uncovering the underlying mechanisms of natural resistance and susceptibility in malaria and the basis of inter-individual variation in immune response to *P. falciparum* infection is a priority to develop more effective vaccines, treatments and public health interventions. In this study, we report the first genome-wide miRNA-mRNA gene expression profiling study of children naturally infected with *P. falciparum*. We used a robust matched design and closely followed a group of children over time from non-infection status, through two main stages of *P. falciparum* infection (asymptomatic and symptomatic parasitemia) and after

treatment. Follow-up profiling allowed for the integration of primary quantitative clinical trait data (parasitemia), mRNA and WGS data to help identify differentially expressed miRNAs associated with parasitemia, under host regulatory genetic control and/or with potential downstream effects on mRNA regulation. While this study design was very challenging to implement in the field, it proved robust to document miRNA early response to *P. falciparum* infection and to identify miRNAs potentially modulating the course of infection.

The observed miRNA expression changes in the early stages of natural *P. falciparum* infection supports a major role of immune cell post-transcriptional regulation in early immune response in blood-stage malaria. First, our data show a coordinated early and fast response as evidenced by the observed gradual up and downregulation of miRNAs expression from the asymptomatic to the symptomatic parasitemia stage. Interestingly, correlation structure of the miRNA transcriptome and expression levels of many differentially expressed miRNAs revert back after treatment to before-infection state. Second, the analysis of temporal in vivo miRNA data together with data and information on their target mRNAs led to the identification of several miRNAs and targets involved in processes key to the pathophysiology of *P. falciparum* infection.

The most significant biological process implicated by integrative miRNA-mRNA and enrichment analyses is lymphocyte cell death. This result is of particular interest given the observed depletion of lymphocytes in our study confirming as well in previous reports[34,35]. Increase in apoptosis of PBMCs has also been described in several protozoan and viral infections[36,37], as well as in mice and individuals infected with *Plasmodium*[38,39]. However, the mechanisms underlying this depletion have not yet been fully understood or systematically investigated in malaria[40]. Our results show that immune cells depletion during *P. falciparum* infection is at least in part mediated by the upregulation of apoptosis-associated miRNAs, in particular miR-15a-5p, miR-16-5p, and miR-181c-5p. In our analysis, these three key miRNAs were identified as upstream regulators targeting several genes involved in apoptosis. We highlight the anti-apoptosis gene BCL2 as one of the key targets of these three miRNAs consistent with our observation of a concurrent decrease in BCL2 expression as miR-15a-p, miR-16-5p and miR-181c-5p expression levels gradually increase during *P. falciparum* infection. Decrease of BCL2 expression in immune cells was reported also in *P. vivax*[41,42] and our results identified miRNAs that are most likely responsible for this.

To independently and functionally validate this finding we experimentally showed that miR-16-5p triggers a decrease in cell

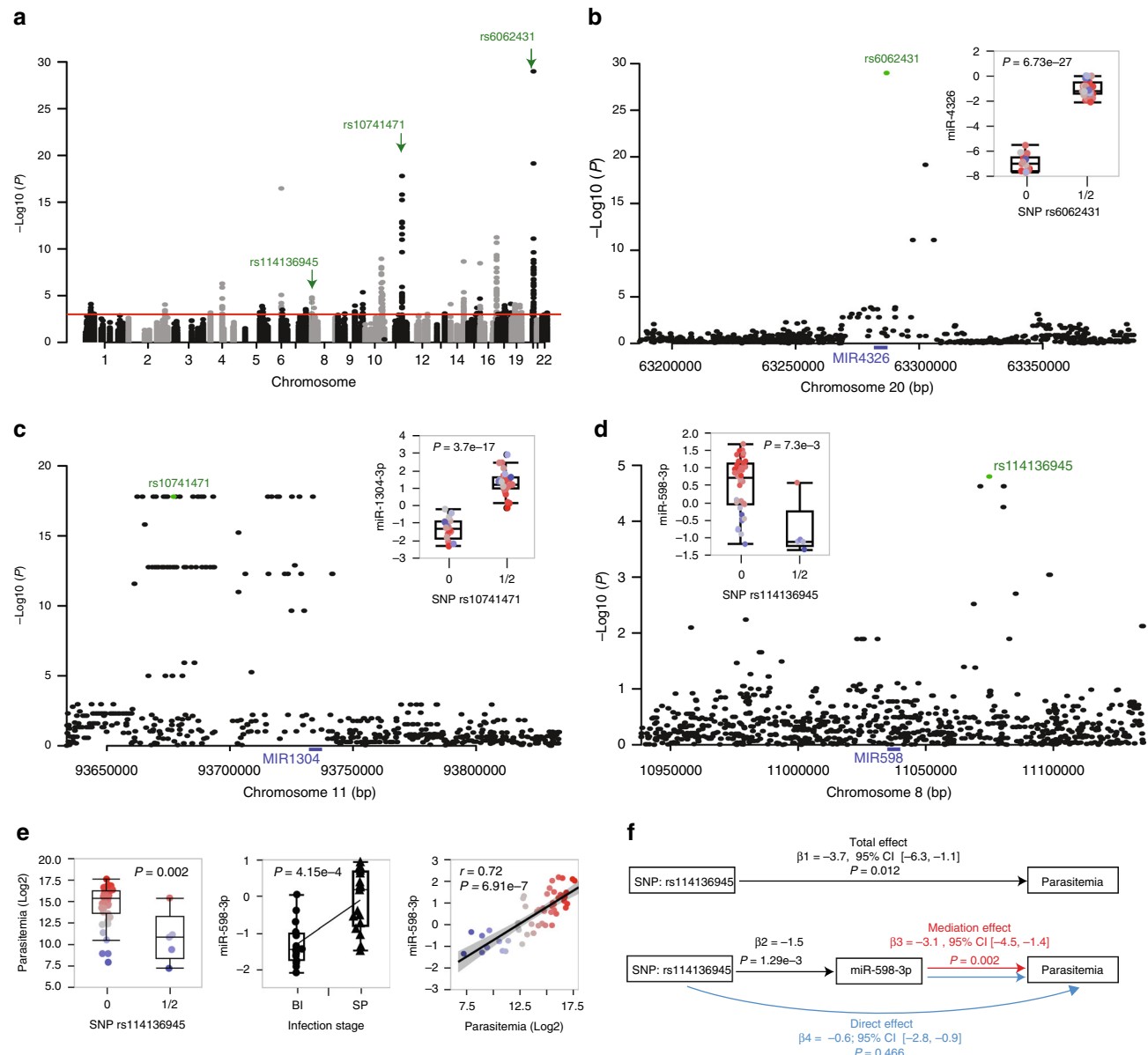

**Fig. 3 Genetic control of miRNA expression in malaria. a** Manhattan plot of all associations tested between 276 miRNAs and 129,310 *cis*-SNPs. Associations implicating miR-4326, miR-1304-3p, and miR-598-3p are shown in green and in **b**–**d**, respectively. The associations shown are for SNPs spanning a 200-kb window centered on the miRNA in question. Peak SNPs are shown in green. The boxplots in **b**–**d** show miRNA expression and associated SNP data with host genotype (0, reference allele homozygote individuals versus 1/2, heterozygotes and alternative allele homozygote individuals (miR-4326/SNP rs6062431, miR-1304-3p/SNP rs10741471, and miR-598-3p/SNP rs114136945, $n = 46$). B–H FDR P values (linear regression model in PLINK v1.9) for each association are shown. **e** Association between parasitemia and SNP rs114136945 (left graph, nominal P value = 0.002, two-tailed student's t test), miR-598-3p expression in the BI ($n = 19$) and SP ($n = 17$) groups (middle graph, B–H $P = 4.15e{-4}$, repeated measure ANCOVA), and right, correlation between miR-598-3p expression and log2 parasitemia in the replication set ($n = 53$, Pearson correlation coefficient $r = 0.72$, and $P = 6.9e{-7}$, multiple regression adjusting for sex, age, and white blood cell count). Each individual is colored by its log2 parasitemia (from the lowest (blue) to the highest (red). The box plots in **b**–**e** show the median, the 25th and 75th percentiles as box edges, and the 5th and 95th percentiles as bounds of whiskers. **f** Mediation analysis to evaluate potential causal relationship between SNP rs114136945 and log2 parasitemia through modulation of miR-598-3p expression. The regression coefficients β, their 95% confidence intervals and P values for each test are shown: $\beta_1$ (−3.7), [−6.3, −1.1], $P = 0.012$, $\beta_2$ (−1.5), $P = 1.29e{-3}$, $\beta_3$, (−3.1), [−4.5, −1.4], $P = 0.002$ and $\beta_4$ (−0.6), [−2.8, −0.9], $P = 0.466$.

survival compared to control transfected cells in two human cell lines. A recent paper reported similar results by showing the deleterious effect of miR-15 and miR-16 on memory T cell differentiation, cell cycle and survival[26]. Another study demonstrated that miR-16 may affect other cell populations of the immune system such as B cells[43]. Combined, our results supported by the findings of these studies support the idea that, upon *P. falciparum* infection, upregulation of miR16-5p is responsible

at least partially for the observed depletion of the lymphocyte pool, a process likely responsible for the defective adaptive lymphocyte responses observed in malaria. This mechanism would prevent or delay acquisition of long-term immunity in malaria through apoptosis of T cells and other lymphocytes.

One important question yet to be addressed experimentally is the blood cell origin of miR16-5p in malarial children. Retrieval of miR-16-5p RNASeq expression data from the BloodmiRs

**Table 1 Top 34 most significant miRNA-SNP associations (Linear regression, PLINK v1.9).**

| miRNA | Chr | SNP | SNP location | A1 | MAF | P value[a] | B-H FDR[b] | Parasitemia? | DE? |
|---|---|---|---|---|---|---|---|---|---|
| miR-4326 | 20 | rs6062431 | Chr20:63286812 | G | 0.47 | 1.01E-29 | 6.73E-27 | 0 | 0 |
| miR-1304-3p | 11 | rs10741471 | Chr11:93677132 | C | 0.36 | 1.58E-18 | 3.68E-17 | 0 | 0 |
| miR-30a-5p | 6 | rs6902769 | Chr6:71412970 | C | 0.47 | 3.38E-17 | 1.71E-14 | 0 | 0 |
| miR-5189-5p | 16 | rs55823018 | Chr16:88468834 | C | 0.35 | 5.82E-12 | 4.47E-09 | 0 | 0 |
| miR-941 | 20 | rs2065993 | Chr20:63914998 | G | 0.39 | 2.33E-10 | 5.36E-09 | 0 | 0 |
| miR-345-5p | 14 | rs12323908 | Chr14:100302297 | T | 0.13 | 2.17E-09 | 2.77E-07 | 0 | 0 |
| miR-1287-5p | 10 | rs7090812 | Chr10:98420452 | A | 0.21 | 1.14E-09 | 8.73E-07 | 0 | 0 |
| miR-3176 | 16 | rs8054514 | Chr16:543277 | T | 0.46 | 3.42E-09 | 1.28E-06 | 0 | 0 |
| miR-4482-3p | 10 | rs1147611 | Chr10:104265500 | G | 0.29 | 3.55E-07 | 2.67E-05 | 0 | 0 |
| miR-1255a | 4 | rs1031034 | Chr4:101302229 | A | 0.20 | 5.11E-07 | 6.81E-05 | 0 | 0 |
| miR-1304-5p | 11 | rs10741471 | Chr11:93677132 | C | 0.36 | 6.63E-06 | 1.54E-04 | 0 | 0 |
| miR-409-3p | 14 | rs2145500 | Chr14:100987907 | T | 0.47 | 4.31E-06 | 1.02E-03 | 0 | 0 |
| miR-624-5p | 14 | rs111287623 | Chr14:31095061 | A | 0.05 | 8.16E-05 | 1.41E-03 | 1 | 1 |
| miR-126-5p | 9 | rs185727058 | Chr9:136639791 | T | 0.05 | 4.35E-06 | 2.95E-03 | 1 | 1 |
| miR-1270 | 19 | rs62107923 | Chr19:20381212 | G | 0.32 | 8.61E-05 | 6.22E-03 | 0 | 0 |
| miR-598-3p | 8 | rs114136945 | Chr8:11074890 | A | 0.08 | 1.59E-05 | 7.22E-03 | 1 | 1 |
| miR-548at-5p | 17 | rs11651671 | Chr17:42494785 | A | 0.14 | 1.08E-04 | 8.41E-03 | 0 | 0 |
| miR-7706 | 15 | rs12437885 | Chr15:85378638 | A | 0.25 | 2.08E-05 | 8.47E-03 | 0 | 1 |
| miR-4781-3p | 1 | rs633883 | Chr1:54138903 | G | 0.10 | 7.85E-05 | 9.65E-03 | 0 | 0 |
| miR-335-3p | 7 | 7_130525724 | Chr7:130525724 | C | 0.05 | 3.97E-04 | 1.81E-02 | 0 | 0 |
| miR-186-5p | 1 | rs74440022 | Chr1:71043652 | T | 0.18 | 8.60E-04 | 2.09E-02 | 0 | 1 |
| miR-375 | 2 | rs6736962 | Chr2:219024931 | G | 0.20 | 8.97E-05 | 2.10E-02 | 0 | 1 |
| miR-4516 | 16 | rs2516726 | Chr16:2045064 | T | 0.38 | 2.54E-04 | 2.46E-02 | 0 | 0 |
| miR-3615 | 17 | rs116362258 | Chr17:74716747 | G | 0.07 | 7.54E-05 | 2.81E-02 | 0 | 0 |
| let-7d-5p | 9 | rs10761319 | Chr9:94143667 | G | 0.27 | 6.47E-04 | 2.96E-02 | 0 | 0 |
| miR-25-3p | 7 | rs11558475 | Chr7:100056977 | G | 0.14 | 1.39E-03 | 3.16E-02 | 0 | 1 |
| miR-629-5p | 15 | rs2927401 | Chr15:70164564 | A | 0.18 | 1.99E-04 | 3.27E-02 | 0 | 0 |
| miR-769-5p | 19 | rs10402427 | Chr19:45922752 | T | 0.32 | 9.95E-05 | 3.38E-02 | 0 | 0 |
| miR-1254 | 10 | rs56051242 | Chr10:68784821 | C | 0.09 | 2.13E-03 | 3.58E-02 | 0 | 0 |
| miR-1273g-3p | 1 | rs72897371 | Chr1:52916937 | G | 0.23 | 3.59E-04 | 4.14E-02 | 0 | 0 |
| miR-378a-3p | 5 | rs73796281 | Chr5:149818327 | C | 0.05 | 1.27E-04 | 4.31E-02 | 1 | 0 |
| miR-874-3p | 5 | rs10058468 | Chr5:137588260 | T | 0.25 | 5.99E-04 | 4.55E-02 | 1 | 0 |
| miR-23b-3p | 9 | rs10993460 | Chr9:95032894 | C | 0.21 | 1.18E-04 | 4.66E-02 | 0 | 0 |
| miR-127-3p | 14 | rs12881727 | Chr14:100956035 | T | 0.43 | 3.47E-04 | 4.90E-02 | 0 | 1 |

Status of miRNA association with parasitemia and of differential expression (DE) between before infection and symptomatic stages is shown (0, no significant association or no DE; 1 significant association or DE).
A1. Minor allele, MAF Minor allele frequency, DE differential expression.
[a]Nominal P value.
[b]Benjamini–Hochberg FDR.

reference dataset[44] of detailed miRNA expression profiles from seven types of human peripheral blood cells (NK cells, B lymphocytes, cytotoxic T lymphocytes, T helper cells, monocytes, neutrophils, and erythrocytes) shows an intriguing enrichment of miR-16-5p expression in erythrocytes, relative to other blood cell types (Supplementary Fig. S4). It is tempting to speculate that lymphocyte depletion is driven by the action of erythrocyte-derived miR-16-5p internalized by lymphocyte cells. A similar mechanism has previously been reported for regulatory Ago2-miRNA complexes in malaria[20]. Addressing this question for miR-16-5p or other miRNAs identified in our study would require blood cell fractionation, isolation of extracellular vesicles and further functional validation experiments.

Understanding the contribution of host genotype to inter-individual variation in miRNA expression traits associated with *P. falciparum* infection was another important aim of our study, in particular given the scarcity of miR-eQTL studies[45–47] and lack of data in the context of natural infection. Our association mapping was performed on all miRNAs detected in blood but our focus was on miRNAs responsive to infection and/or associated with parasitemia. The possibility that infection or parasitemia-associated miRNA response is dependent on host regulatory genetic is particularly relevant in the context of mapping the genetic basis of susceptibility to malaria and personalized medicine[48]. Our results revealed that expression levels of several

miRNAs are under host genetic regulatory control supporting the hypothesis that host regulatory variation is one of the factors influencing the course of *P. falciparum* infection. In line with this, we report seven miRNAs (miR-126-5p, miR-3201, miR-378a-3p, miR-510-3p, miR-598-3p, miR-624-5p, miR-7706) responsive to infection and/or correlated with parasitemia that are also under genome-wide significant host genetic control. Of these we highlight in particular the association between miR-598-3p and SNP rs114136945 given supportive evidence from mediation analysis for a causative protective effect of the minor allele on parasitemia. Further functional tests are required to reveal the cell-type context of such effect as well as the downstream intermediate cellular processes leading to control parasite proliferation and subsequently decreased parasitemia. It remains also to identify if the other identified *cis*-miRNAs are cell type-specific or not. This will require innovative approaches such as single-cell genomic profiling of PBMCs[49,50] given that cell type fractionation techniques are extremely challenging to implement in fieldwork settings in Africa.

In conclusion, our findings highlight post-transcriptional regulation as a potential key molecular mechanism that impacts immune response to *P. falciparum* infection. These regulatory effects are likely mediated by modulation of the expression levels of miRNAs and mechanisms promoting immune evasion through depletion of key lymphocyte cell populations. We also

demonstrate a role of host genetic regulatory variation in controlling expression of miRNAs associated with parasitemia providing evidence for a major source of inter-individual variation in host immune response to *P. falciparum* infection. The results demonstrate also the power of matched/paired designs with close follow up of study participants and the use of integrative genomic approaches and endophenotypes to help overcome the challenges faced with traditional genetic mapping approaches in Africa. We call for using this approach in settings where obtaining large sample sizes is challenging and where controlling for variation due to environmental factors and distinguishing causal and response effects are needed.

## Methods

**Data and sample collection protocol**. The study was approved by the Ethical Committee of the Ministry of Health of Burkina Faso (Ministry of Health, Burkina Faso; protocol number 2015-02-018) and the Institutional Review Board of New York University Abu Dhabi (UAE, protocol number 011-2015). Informed consent was obtained from the children and/or parents following the approved protocols. Blood samples (~10 ml) and patient data were collected by a trained team at the CNRFP clinic in Banfora. The same protocol was used for all samples to minimize heterogeneity due to technical reasons. All samples were collected between 9:00 a.m. and 2:00 p.m. and processed immediately after collection. Three mls of blood were collected in Tempus tubes (ThermoFisher) to stabilize RNA. The remaining blood was used for standard thick blood smear analysis, to generate total cell counts and hemoglobin levels as well as for DNA extraction. Hemoglobin and G6PD genotypes were determined as previously described[51,52]. Characteristics of the study participants, biochemical and cellular data are summarized in Supplementary Data 1.

**miRNAs expression profiling**. Whole blood RNA was isolated using Tempus Spin RNA isolation kit (Thermo Fisher). Qubit and a 2100 Bioanalyzer instruments were used for RNA quantification and quality control. Small RNA libraries were prepared using TruSeq Small RNA kit (Illumina) and 1 μg of high-quality total RNA (RNA Integrity Number RIN > 8). Library size distribution was checked using a 2100 Bioanalyzer instrument. Throughout the experimental protocols, careful attention was paid to minimize batch effects. Importantly, the samples and libraries were randomized and processed the same way from total RNA extraction and library preparation, to the sequencing. Libraries were sequenced on 12 lanes of HiSeq Rapid kit (50 cycles) using the HiSeq 2500 instrument (Illumina) at New York University Abu Dhabi.

**Total RNA expression profiling and DNA sequencing**. Total RNA sequencing data was generated from the same stock of total RNA samples used for miRNA profiling. For each sample, 1 μg of total RNA was subject to globin-mRNA depletion using the GlobinClear kit (Thermo Fisher) followed by library preparation using 500 ng of high-quality globin-depleted RNA (RIN > 8) and KAPA Stranded RNA-Seq kit with RiboErase (Kapa Biosystems). The libraries were quality-checked and quantified prior to multiplexing and 100 bp paired-end sequencing using the TruSeq SBS Kit v3 - HS (Illumina) on a HiSeq 2500 instrument (Illumina). Genomic DNA libraries were constructed using the TruSeq Nano DNA Library Prep Kit (Illumina) and sequenced at 30× coverage on the HiSeq X platform (Illumina) at the Australian Genome Research Facility in Melbourne, Australia.

**Bioinformatics analyses of RNA sequencing data**. For miRNA data, raw sequencing reads were demultiplexed using standard Illumina pipeline with bcl2fastq and processed with Trimmomatic v0.36 to remove indexes and adapters. Trimmed reads were further processed with the FASTX Toolkit v0.0.14 to filter out reads with tail quality <15 nucleotides and retain reads of 16–25 nucleotides length for downstream analyses. FastQC v0.11.5 was used to visualize quality metrics before and after filtering and high-quality reads were subject to small RNAs annotation and quantification using the online tool OASIS 2.0[53,54] and miRbase v21 (Supplementary Data 2). miRNAs expressed at a minimum count of 10 reads in at least 50% of the samples per experimental condition were retained, producing a final dataset of 320 miRNAs in the discovery and replication sets. For the total RNA data, raw reads were processed using Trimmomatic v0.36 to remove adapter sequences and low-quality bases (Phred quality score <20). Filtered reads were then mapped to the human reference genome (Ensembl GRCh38 release-84) using STAR v2.5.0c. Transcripts abundance (Fragments Per Kilobase of transcript per Million) was estimated using cufflinks v2.2.1 and converted to transcripts per million (TPM) using a custom script (https://github.com/Yidaghdour/malaria-miRNA). Only genes with a minimum of 1 TPM in 10 samples per experimental condition were retained for downstream analyses resulting in 12,375 genes.

**Bioinformatics analyses of DNA sequencing data**. Sequencing reads were analyzed following GATK Best Practices for variant discovery analysis. Briefly, alignments of quality-trimmed reads were performed using Bowtie 2 v2.2.8 and the GRCh38 human reference sequence. Local alignments were generated and base quality scores calibrated using GATK 3.5-0. gVCF files were annotated using SnpEff-4.3.2. Variants with coverage >10×, GQ score >20, call rate >90, minor allele frequency (MAF) > 0.05 and in Hardy-Weinberg equilibrium (HWE, $P <$ 0.05) were retained for association mapping analysis.

**miRNA and total RNA data normalization**. miRNAs and mRNA raw counts were log2 transformed, before being normalized using JMP Genomics 8 (SAS Institute). For miRNA, only data from the discovery set was first batch normalized (correcting for the effect of library preparation done in three sets of randomized samples) to further reduce any potential technical batch effects. Then, mean normalization was applied on both sets (replication and discovery). Based on distribution and unsupervised analyses of the log-scaled data before and after normalization four outlier samples were removed resulting in a final dataset of 68 (discovery set) and 53 (replication set) miRNA samples showing consistent distribution (Supplementary Fig. S1). For total RNA, log10-scaled TPM data was subject to similar quality control analysis followed by IQR normalization using JMP Genomics 8 (SAS Institute). We tested four normalization methods (mean, median, IQR and quantile normalization) and all gave similar results in both unsupervised and supervised analysis.

**miRNAs unsupervised and supervised analyses**. Unsupervised and supervised analyses were performed on normalized gene expression data using JMP Genomics 8 (SAS Institute). The magnitude and significance of differential expression of individual miRNA were evaluated by repeated measures analysis of covariance (ANCOVA) to account for the individual effect using the following model where $\mu$ is the mean measure of transcript abundance and the error $\varepsilon$ is assumed to be normally distributed with a mean of zero:

miRNA expression $= \mu +$ Age + Sex + WBC + Individual effect + Infection status $+ \varepsilon$ (discovery set).

Association between miRNA expression and parasitemia was assessed using the following multiple regression model:

miRNA expression $= \mu +$ Age + Sex + WBC + Parasitemia $+ \varepsilon$ (replication set).

Benjamini-Hochberg FDR of 5% was used as a threshold of statistical significance in both analyses.

**Predictive modeling**. Predictive modeling was performed to test the power of miRNA traits to predict parasitemia using the predictive modeling workflow implemented in SAS (SAS Institute). Predictor reduction was based on a B-H FDR threshold of 5% of Pearson correlation using a stepwise general linear model accounting for age, sex and white blood counts and including all 72 miRNAs associated with parasitemia. Model selection was based on the Akaike information criterion. A set of four miRNAs were identified as the best predictive variables (hsa-miR-199-3p, hsa-miR-3173-5p, hsa-miR-342-3p, and hsa-miR-532-5p). Twofold cross-validation was done by randomly shuffling the dataset into training and test sets (60 and 40% of the full dataset, respectively). Three metrics were used to assess the performance of the model for both training and test sets. First, the Root Mean Squared Error (RMSE) of the model. A lower RMSE value indicates a better fit. Second, Harrell's C-Statistic was used to estimate the probability of concordance between predicted and observed parasitemia values. Harell's C-Statistic values range between 0.5 and 1 and a value of 0.5 indicates a total disagreement between predicted and observed parasitemia values. Third, Pearson correlation between predicted and observed parasitemia values. High Pearson correlation values indicate a good performance of the model.

**Integrative, pathway enrichment, and cell type proportion analyses**. mRNA transcripts expressed during symptomatic parasitemia in the replication set (12,375 transcripts) were cross-correlated with 133 miRNAs that are differentially expressed after infection with *P. falciparum* and/or associated with parasitemia. Subsequently, statistically significant ($P <$ 0.05) anti-correlated miRNA-mRNA pairs were filtered and their interaction analyzed based on prediction (high prediction score) and/or experimental observation using the MicroRNA Target Filter Tool (Ingenuity Pathway Analysis, IPA). The resulting list of targets was then used for pathway, molecular and cellular functions enrichment using the Benjamini–Hochberg procedure to correct for multiple testing. The Epic deconvolution tool[31] was used to estimate the proportions and types of circulating immune cells in each sample using total RNA data.

**qPCR validation of miRNAs**. To confirm the expression of 16 miRNAs (miR-1260b, miR-148b-5p, miR-150-3p, miR-16-5p, miR-20a-5p, miR-25-5p, miR-29a-3p, miR-30e-3p, miR-3195, miR-451a, miR-548k, miR-598-3p, miR-664a-3p, miR-6803-3p, miR-93-3p, miR-98-5p) differentially expressed between symptomatic parasitemia and before infection samples, we performed qPCR using 39 samples for miRNAs. All samples were selected from our larger cohort for which total RNA was available to perform reverse transcription using miScript II RT (Qiagen) for

miRNAs. The primers for miRNAs and the housekeeping control (SNORD26) were obtained from Qiagen (Supplementary Data 4). Quantitative-PCR of the miRNAs were performed using miScript SYBR Green PCR Master Mix Kit and the BioMark HD instrument (Fluidigm). All qPCR reactions were performed in duplicate. Relative expression levels measured as threshold cycle value (Ct) were normalized and then analyzed using Qiagen data analysis center (GeneGlobe, Qiagen, https://geneglobe.qiagen.com/us/analyze/) and JMP Genomics 8.

**Experimental analysis of cell survival using mimic miR-16-5p expression**. HEK and Hela cell lines grown to 80–90% confluence with DMEM, 10% FBS were plated overnight at 1,000–3,000 cells in a total volume of 200 μl per well using a 96-well flat-bottom. The following day, the cells were transfected with 10 pM of either miR-16-5p mimic or a negative control (Sigma) using Lipofectamin RNAiMax (Thermo Scientific). Forty-eight hours post transfection, 20 μL of MTT (5 mg/mL) in PBS was added to each well and incubated for an additional 2 h at 37 °C. After the incubation, the supernatant was removed and the formazan crystals solubilized in 100 μL of DMSO, and the absorbance read at 570 nm using a microplate reader (Biotec). Six independent treatments were performed ($n = 6$, with 2–4 technical replicates per treatment).

**miRNA-eQTL analysis**. Expression quantitative trait loci (eQTL) analysis was performed on SNPs genotypes and miRNAs expression levels using an additive multiple regression model in PLINK v1.9. Given that genetic regulatory effects on miRNA expression are predominantly proximal to miRNA (25, 26) and the limited statistical power to detect distal- and trans-regulatory effects, we focused our eQTL analysis on cis-acting regulatory effects using the replication dataset. For each miRNA in our replication dataset, the level of expression was tested against all variants (MAF > 5%, HWE $P > 0.05$) located within a window of 200 kb centered from the miRNA. The entire allelic dataset was coded as 0 and 1 where each number represents the number of copies of the minor allele. In total, 129,310 cis-SNPs and 276 miRNAs were included corresponding to an average of 468 SNPs per miRNA. The following model was used to test for miRNA-SNP associations using B-H FDR and 100,000 permutations to assess statistical significance:

$$\text{miRNA expression} = \mu + \text{SNP} + \text{Age} + \text{Sex} + \text{WBC} + \text{Parasitemia} + \varepsilon$$

**Mediation analysis**. Mediation analysis was performed using multiple linear regression models as implemented in the R package Mediation[55]. This analysis was restricted to the miRNAs under genetic control and that are significantly associated with parasitemia. One outlier data value was removed prior to performing the test. The bootstrap $P$ value using 1,000 simulations was used to assess statistical significance of the mediation effect ($\beta_3$). Estimates and $P$ values of the total and direct effect of genetic variant on parasitemia ($\beta_1$ and $\beta_4$, respectively), the effect of the genetic variant on miRNA expression ($\beta_2$) were assessed and reported.

**Reporting summary**. Further information on research design is available in the Nature Research Reporting Summary linked to this article.

## Data availability

All the miRNAs and mRNA expression data reported in this paper was deposited in the Gene Expression Omnibus (GEO) database under accession number GSE144486. The genotyping dataset generated and analyzed during the current study are available from the corresponding author on reasonable request. Access to the data will be granted to researchers for appropriate use consistent with the consent provided by the study participants.

## Code availability

Code used in the analysis is hosted at https://github.com/Yidaghdour/malaria-miRNA[56].

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

## Acknowledgements

We would like to thank the children who participated in the study and their families. We thank all the individuals who facilitated sample collection and clinical work in particular Guigma Telesphore, Ouattara Aristide and the staff of the Centre National de Recherche et de Formation sur le Paludisme. We thank Wael Abdrabou for assisting with figures and various aspects of the malaria research program. We thank Kurt Warren, Ibrahim Al Hawai, Reza Rowshan, Kris Gunsalus, the NYUAD Core Bioinformatics and Technology Platform teams, Mehar Sultana and Raneem Eteer for assistance with logistics and experiments. We thank Greg Gibson and Jane Carlton for helpful discussions and suggestions. This work is supported by NYUAD Grant AD105 to Youssef Idaghdour.

## Author contributions

Y.I., A.Diawara, M.M.D., A.B.T., and I.S. designed the study. A.Diawara and M.M.D. coordinated the project. I.S. and S.B.S. supervised fieldwork. S.A.C. and A.B. performed clinical work. M.M.D., A.Diawara, S.S., S.S.S., and A.Diarra collected the samples and laboratory data. A.Diawara, M.M.D., M.A., and H.T.E.J. performed genomic experiments. V.M., N.D., and A.Y. performed bioinformatic analysis. A.Diawara, M.M.D., and Y.I. analyzed and interpreted the data and wrote the manuscript. All authors read and approved the manuscript.

## Competing interests

The authors declare no competing interests.
