## [Peer Review File · Nature Communications]

Reviewers' Comments:

Reviewer #1:

Remarks to the Author:

Review for the Dieng et al.

The current study reports the genetic analysis of blood microRNA, mRNA and SNPs analysis of a longitudinal cohorts of children during the clinical course of malaria development. They collected a discovery dataset, which is followed up by the validation cohorts. Through the integrative analysis, they identified 36 microRNAs associated with the infection and severity of parasitemia. These microRNAs are enriched in their targeting of the BCL-2 and other apoptotic genes. The strength of the manuscript is the longitudinal follow-up of the two cohorts with detailed molecular and phenotypic analysis. The associated blood microRNAs and mRNAs provide novel biological insights into the host post-transcriptional regulation of *P. falciparum*.

Some concerns include:

1. Sickle cell trait would be definitely a potential confounding factor. Individuals with sickle cell traits won't develop symptoms. I am a bit surprised and disappointed that the authors did not exclude this possibility using molecular analysis. Instead, they just state that the recruit cohorts with no disease symptoms or chronic disease history (including sickle cell disease).
2. The red cells are the host cell for *P. falciparum*. Red cells have abundant level of the microRNAs and may be a significant contributor to the whole blood microRNAs. Some of the cited references are actually related to red cell microRNAs. The authors need to discuss the possibly of red cell RNA and determined the likely sources of the elevated miR-16-5p in their datasets since this is critical in the hypothetic mechanisms.
3. The idea of the lymphocyte depletion associated with the three apoptosis-related microRNA is interesting. However, do the expression of these microRNAs associate with varying degree of lymphocyte depletion ?
4. First part of the manuscript focuses on the association between microRNA-mRNA that converges on the apoptotic processes. Second part of the manuscript describes SNP and microRNA expression. The particular association of miR-598-3p with SNP rs114136945 and parasitemia is quite interesting. But these two parts of the manuscripts seems unrelated and disjointed.

Reviewer #2:

Remarks to the Author:

The manuscript by Dieng and colleagues investigates miRNAs in a longitudinal cohort of children before and after infection with *P. falciparum*. Malaria is a major cause of morbidity and mortality that affects predominantly Sub-Saharan Africa. The biological mechanisms underlying variation in host immune response to *P. falciparum* infection are poorly understood. This manuscript focuses on host gene expression response to *P. falciparum*, with a specific focus on miRNAs and host genetics. This is a very exciting manuscript with a clever study design that includes a longitudinal discovery cohort to study response to infection, and a replication cohort to study parasitemia. Additionally, the study of miRNA and mRNA regulation mediated by miRNA is a promising yet underdeveloped field in host immune response to pathogens. Another strength of the manuscript is the in vitro study to confirm that miR-16-5p regulation of the response to infection leads to changes in cell proliferation.

Despite these strengths, there are a few aspects of the manuscript that require careful attention. In general the statistical analyses are not well described, therefore making it hard to judge their rigor. Detailed comments are below.

Major comments:

1. The following section is unclear and additional details on this set of analyses should be provided. Which model was used? For each metric, which thresholds were used?
"To test the predictive power of miRNA expression profiles, expression levels of miRNAs associated with parasitemia in the replication set were fitted to a linear regression using a stepwise general linear model (GLM). Several metrics such as the Root Mean Squared Error (RMSE), the Harrell's concordance index (Harrell's C) and correlation coefficient (r) were used to assess the performance of the model. Significance threshold was set as FDR 5%."
2. Which background was used for enrichment analysis? All expressed miRNA/genes should be used not a general genomic background. Please clarify if the results of this analysis are multiple-test corrected. If this is not the case, please report only multiple test corrected enrichment results or remove this section from the manuscript.
3. Genetic analysis. Please add more details on the permutations. Were they run on all possible SNP-miRNA pair or only on the top signal per locus? Please report significant findings at a given FDR threshold (e.g. 10%) rather than using a permutation p-value threshold, as this does not take into account multiple test correction. Please include a qq-plot of the p-values after correction, to confirm that they follow a uniform distribution.
5. This dataset should be amenable to perform response (infection) miR-eQTL mapping, thus directly investigating genetic variants that affects response to infection. Are there genetic variants that regulate miRNA in response to infection in these samples?

Minor comments:

1. In figure 1c and 1h, the lines are distracting, adding a dot to represent the median and also marking the interquartile range should communicate the same trend with increased clarity.
2. Line 271: "FDR-adjusted P-value = $6.91e-7$ ", is this the p-adjusted following the Benjamini-Hochberg procedure? Or is this the q-value? Please clarify.

REVIEWER COMMENTS

Reviewer #1 (Remarks to the Author):

Review for the Dieng et al.

The current study reports the genetic analysis of blood microRNA, mRNA and SNPs analysis of a longitudinal cohorts of children during the clinical course of malaria development. They collected a discovery dataset, which is followed up by the validation cohorts. Through the integrative analysis, they identified 36 microRNAs associated with the infection and severity of parasitemia. These microRNAs are enriched in their targeting of the BCL-2 and other apoptotic genes. The strength of the manuscript is the longitudinal follow-up of the two cohorts with detailed molecular and phenotypic analysis. The associated blood microRNAs and mRNAs provide novel biological insights into the host post-transcriptional regulation of *P. falciparum*.

Some concerns include:

1. Sickle cell trait would be definitely a potential confounding factor. Individuals with sickle cell traits won't develop symptoms. I am a bit surprised and disappointed that the authors did not exclude this possibility using molecular analysis. Instead, they just state that the recruit cohorts with no disease symptoms or chronic disease history (including sickle cell disease).

Reply: We agree with the reviewer that sickle cell trait would be definitely a potential confounding factor. In fact, we have used molecular analysis to confirm initial information about the disease (initially obtained from medical records of the children at the CNRFP). In our initial submission, we mentioned in the methods section that hemoglobin genotypes were generated and we have provided them in Supplementary Table 1 but failed to mention this in the results section. For carriers of the trait, we excluded as many individuals as we could and only three carrier children (genotype AS) were included in the study. In the entire study cohort (discovery and replication combined), 88% and 8% of the children have AA and AC genotype, respectively. In our PCA, AS/AC individuals are not outliers and do not cluster together. Furthermore, in the replication cohort, levels of parasitemia are not significantly different (t-test, p-value = 0.8) between the six AS/AC children and AA children (equal group size and randomly selected among the AA children matched for sex and age). We now mention this by including the following statement in the methods, page 18, lines 484-490:

“Since sickle cell disease and traits have protective effects on malaria and to avoid potential confounding effects, children with the disease were not included in the study and every effort was made to limit the number of children with the trait. In the discovery and replication cohorts combined, 88% of the children have a hemoglobin AA genotype and in the replication cohort six hemoglobin AC/AS children cluster randomly in PCA and their parasitemia levels were not statistically different relative to a sex and age-matched AA hemoglobin genotype group (Student's t-test, P-value = 0.8).”

2. The red cells are the host cell for *P. falciparum*. Red cells have abundant level of the microRNAs and may be a significant contributor to the whole blood microRNAs. Some of the cited references are actually related to red cell microRNAs. The authors need to discuss the possibly of red cell RNA and determined the likely sources of the elevated miR-16-5p in their datasets since this is critical in the hypothetic mechanisms.

Reply: This is indeed an important point. Our dataset is generated from whole blood and therefore it is not possible for us to know the cell type(s) of origin miR-16-5p expression. However, retrieval of miR-16-5p RNASeq expression data from a reference dataset (BloodmiRs) of detailed miRNA expression profiles from seven types of human peripheral blood cells (NK cells, B lymphocytes, cytotoxic T lymphocytes, T helper cells, monocytes, neutrophils and erythrocytes) shows an intriguing enrichment of miR-16-5p expression in erythrocytes (CD235a), relative to other blood cell types (see the figure below). Nonetheless, miR-16-5p is expressed in all major white blood cells. Without cell type fractionation and profiling, it would therefore be extremely hard to infer if cell type-specific elevation is taking place or if erythrocyte-derived miR-16-5p internalized by lymphocyte cells are responsible for the observed depletion. BCL2 on the other hand is expressed mainly in B and T cells (see the figure below) and therefore it is plausible that the apoptotic effect of miR-16-5p via BCL2 is taking place in these cells as supported by our analysis. Although it is extremely challenging to conduct these experiments in African settings, it is hoped that in the future we will be able to fractionate the cells in the field and subsequently perform cell type-specific profiling to adequately answer this question.

miR-16-5p RNASeq expression data retrieved from the BloodmiRs reference dataset of detailed miRNA expression profiles from seven types of human peripheral blood cells (NK cells, B lymphocytes, cytotoxic T lymphocytes, T helper cells, monocytes, neutrophils and erythrocytes).

<https://www.ncbi.nlm.nih.gov/pmc/articles/PMC5766192/>

<http://134.245.63.235/ikmb-tools/bloodmiRs/>

Consensus normalized expression levels of BCL2 gene for 6 blood cell types, created by combining the data from the three transcriptomics datasets (HPA, GTEx and FANTOM5, data retrieved from <https://www.proteinatlas.org/ENSG00000171791-BCL2/tissue>).

To address this point, we have now added the following statement in the discussion in page 12, lines 346-356:

“One important question yet to be addressed experimentally is the blood cell origin of miR16-5p in malarial children. Retrieval of miR-16-5p RNASeq expression data from the BloodmiRs reference dataset (ref. 44) of detailed miRNA expression profiles from seven types of human peripheral blood cells (NK cells, B lymphocytes, cytotoxic T lymphocytes, T helper cells, monocytes, neutrophils and erythrocytes) shows an intriguing enrichment of miR-16-5p expression in erythrocytes, relative to other blood cell types (Supplementary Figure S4). It is tempting to speculate that lymphocyte depletion is driven by the action of erythrocyte-derived miR-16-5p internalized by lymphocyte cells. A similar mechanism has previously been reported for regulatory Ago2-miRNA complexes in malaria (ref. 20). Addressing this question for miR-16-5p or other miRNAs identified in our study would require blood cell fractionation, isolation of extracellular vesicles and further functional validation experiments.”

3. The idea of the lymphocyte depletion associated with the three apoptosis-related microRNA is interesting. However, do the expression of these microRNAs associate with varying degree of lymphocyte depletion?

Reply: The three apoptosis-related microRNA (miR-15a-5p, miR-16-5p and miR-181c-5p) show a negative correlation with lymphocyte counts in our replication cohort although only the association implicating miR-181c-5p is statistically significant (P -value = 0.02). Leukocyte count is a complex trait and given that (i) we are not accounting for the effects of protein-coding genes that might be associated with lymphocyte counts and (ii) we are profiling whole blood and other cell types might be expressing these miRNAs, the effect we are observing might be weakened by these factors and this is another reason warranting cell type-specific profiling in the future.

4. First part of the manuscript focuses on the association between microRNA-mRNA that converges on the apoptotic processes. Second part of the manuscript describes SNP and microRNA expression. The particular association of miR-598-3p with SNP rs114136945 and parasitemia is quite interesting. But these two parts of the manuscripts seems unrelated and disjointed.

Reply: To make the connection between the two parts clearer, we added in the Results and Discussion sections the following statements as a transition between the two parts:

Page 8, lines 234-238:

“In summary, the analyses detailed above provide evidence for a central role of post-transcriptional regulation in early transcriptional response to natural *P. falciparum* infection. A fundamental question

that follows pertains to the nature and extent of the contribution of regulatory genetic variation to the observed response. To address this question, we set to examine the association between allelic genetic variation and miRNA transcript abundance”

Page 12, lines 361-365:

“Our association mapping was performed on all miRNAs detected in blood but our focus was on miRNAs responsive to infection and/or associated with parasitemia. The possibility that infection or parasitemia-associated miRNA response is dependent on host regulatory genetic is particularly relevant in the context of mapping the genetic basis of susceptibility to malaria and personalized medicine”

Reviewer #2 (Remarks to the Author):

The manuscript by Dieng and colleagues investigates miRNAs in a longitudinal cohort of children before and after infection with *P. falciparum*. Malaria is a major cause of morbidity and mortality that affects predominantly Sub-Saharan Africa. The biological mechanisms underlying variation in host immune response to *P. falciparum* infection are poorly understood. This manuscript focuses on host gene expression response to *P. falciparum*, with a specific focus on miRNAs and host genetics. This is a very exciting manuscript with a clever study design that includes a longitudinal discovery cohort to study response to infection, and a replication cohort to study parasitemia. Additionally, the study of miRNA and mRNA regulation mediated by miRNA is a promising yet underdeveloped field in host immune response to pathogens. Another strength of the manuscript is the in vitro study to confirm that miR-16-5p regulation of the response to infection leads to changes in cell proliferation.

Despite these strengths, there are a few aspects of the manuscript that require careful attention. In general the statistical analyses are not well described, therefore making it hard to judge their rigor. Detailed comments are below.

Major comments:

1. The following section is unclear and additional details on this set of analyses should be provided.

Which model was used? For each metric, which thresholds were used?

“To test the predictive power of miRNA expression profiles, expression levels of miRNAs associated with parasitemia in the replication set were fitted to a linear regression using a stepwise general linear model (GLM). Several metrics such as the Root Mean Squared Error (RMSE), the Harrell’s concordance index (Harrell’s C) and correlation coefficient (r) were used to assess the performance of the model. Significance threshold was set as FDR 5%.”

Reply: We acknowledge that some relevant details were not provided, and so to improve these descriptions, we have now updated the methods section and Supplementary Figure S3 to include this information. The section on page 21, lines 568-583 is updated as follows:

“Predictive modelling was performed to test the power of miRNA traits to predict parasitemia using the predictive modeling workflow implemented in SAS (SAS Institute). First, predictor reduction was

performed by fitting expression levels of 72 miRNAs associated with parasitemia in the replication to a linear regression model accounting for age, sex and white blood counts ($\text{Log}_2 \text{Parasitemia} = \text{miRNA expression} + \text{Age} + \text{Sex} + \text{WBC}$). Predictor reduction was based on a B-H FDR threshold of 5% of Pearson correlation. Next, a set of the best predictive variables (miR-1227-3p, miR-25-5p, miR-30e-3p, miR-3173, miR-342-3p, miR-4775, miR-532-5p) was identified using a stepwise general linear model. Model selection was based on the Akaike information criterion. Three metrics were used to assess the performance of the model. First, the Root Mean Squared Error (RMSE) of the model. A lower RMSE value indicates a better fit. Second, Harrell's C-Statistic was used to estimate the probability of concordance between predicted and observed parasitemia values. Harrell's C-Statistic values range between 0.5 and 1 and a value of 0.5 indicates a total disagreement between predicted and observed parasitemia values. Third, Pearson correlation between predicted and observed parasitemia values. High Pearson correlation values indicate a good performance of the model."

We have also added code used in various analyses reported in the paper and provide the link in a Code Availability statement (<https://github.com/Yidaghdour/malaria-miRNA>).

2. Which background was used for enrichment analysis? All expressed miRNA/genes should be used not a general genomic background. Please clarify if the results of this analysis are multiple-test corrected. If this is not the case, please report only multiple test corrected enrichment results or remove this section from the manuscript.

Reply: We confirm that we used the list of expressed miRNA/genes and not general genomic background. We also confirm that the Benjamini-Hochberg method was used for multiple testing in the disease and molecular functions analysis as implemented in the IPA tool. The results of this analysis were our focus in the paper. Canonical pathways analysis on the other hand was less powered and significance was determined using a right-tailed Fisher's Exact test as recommended by IPA. As requested, we removed the following statement referring to this minor analysis in the main text:

"This analysis implicates miRNA differentiation in the divergence in key cell death and adaptive immune functions including Ceramide Signaling (P-value = 3.48e-03), IL-8 Signaling (P-value = 3.48e-03), T Cell Receptor Signaling (P-value = 4.83e-03) and Role of Cytokines in Mediating Communication Between Immune Cells (P-value = 5.20e-03)."

3. Genetic analysis. Please add more details on the permutations. Were they run on all possible SNP-miRNA pair or only on the top signal per locus? Please report significant findings at a given FDR threshold (e.g. 10%) rather than using a permutation p-value threshold, as this does not take into account multiple test correction. Please include a qq-plot of the p-values after correction, to confirm that they follow a uniform distribution.

Reply: We confirm that permutations were run on all possible SNP-miRNA pairs. We also confirm that we are reporting both B-H FDR and permuted p-values (Table 1 and Supplementary Table S8). At 5% B-

H FDR, the same number of significant eQTL-miRNAs associations was detected (34 miRNAs, 27 of which overlap with those significant in the permutation analysis). At 10% B-H FDR, 49 miRNA eQTL-miRNAs associations were detected. We report the results of both methods in the results section (page 9, lines 252-259) with the details provided in Supplementary Table S8:

“In total, we identified 486 cis-miR-eQTLs for 34 miRNAs (permuted P-value < 0.05, 100,000 permutations (Figure 3a). Table 1 shows a list of 34 significant cis-miR-eQTLs associations including five associations implicating miRNAs differentially expressed between infection stages in the discovery set and/or associated with parasitemia in the replication set. The full list of the 34 significant cis-miR-eQTLs associations is provided in Supplementary Table S8. At 5% B-H FDR, the same number of miRNAs with cis-miR-eQTLs was detected (n = 34 miRNAs, 27 of which overlap with the 34 significant miRNAs in the permutation analysis, Supplementary Table S8).”

As requested, we also include QQ-plots for a random selection of 20 miRNAs listed in Table 1 (New Supplementary Figure S5). The plots confirm the uniform distribution of the values.

5. This dataset should be amenable to perform response (infection) miR-eQTL mapping, thus directly investigating genetic variants that affects response to infection. Are there genetic variants that regulate miRNA in response to infection in these samples?

Reply: This is indeed an excellent suggestion and we have contemplated back-and-forth if we should include this analysis in this paper or not. Initially, we were mainly concerned that our dataset is underpowered for this analysis given that robust detection of interactions requires larger sample sizes. This was indeed confirmed as we only detected two borderline significant response eQTLs. We therefore prefer not to include this analysis in this paper.

Minor comments:

1. In figure 1c and 1h, the lines are distracting, adding a dot to represent the median and also marking the interquartile range should communicate the same trend with increased clarity.

We made these changes as suggested.

2. Line 271: “FDR-adjusted P-value = 6.91e-7”, is this the p-adjusted following the Benjamini-Hochberg procedure? Or is this the q-value? Please clarify.

We confirm that this is a Benjamini-Hochberg adjusted p-value and now clearly state it in the text.

Reviewers' Comments:

Reviewer #1:

Remarks to the Author:

The authors have answered all my questions. I don't have further concerns.

Reviewer #2:

Remarks to the Author:

In this revised version of the manuscript, the authors have addressed most of this reviewer's comments. There are only two standing issues that remain to be addressed:

1. Predictive modeling. The added information on the methods does not explain how the data were handled to obtain separate discovery and validation sets. If an independent validation set was not used and the full dataset was used for both discovery and validation, the analysis should be repeated with appropriate cross-validation.
2. The nominally significant (permutation corrected) p-value from the miR-eQTL analysis should be moved to the supplements and the results section should focus on the FDR-corrected findings.

Response to REVIEWER COMMENTS

We thank the reviewers for their time and helpful feedback that have helped us to improve the manuscript. We have now performed cross-validation and made minor edits to the manuscript.

Reviewer #1 (Remarks to the Author):

The authors have answered all my questions. I don't have further concerns.

Reviewer #2 (Remarks to the Author):

In this revised version of the manuscript, the authors have addressed most of this reviewer's comments. There are only two standing issues that remain to be addressed:

1. Predictive modeling. The added information on the methods does not explain how the data were handled to obtain separate discovery and validation sets. If an independent validation set was not used and the full dataset was used for both discovery and validation, the analysis should be repeated with appropriate cross-validation.

Reply: We appreciate the suggestion and now perform 2-fold cross-validation by randomly shuffling the dataset into discovery and validation sets (60 and 40% of the full dataset, respectively, as indicated in Supplementary Table 1), and assessing performance of the model. The model performed generally well with only four predictor miRNAs (hsa-miR-199-3p, hsa-miR-3173-5p, hsa-miR-342-3p and hsa-miR-532-5p) as shown in the true vs predicted parasitemia plots for the training and test sets:

We have now added the following statements to the methods and results sections and updated Supplementary Figure S3 showing the parameters and results of the analysis:

Methods:

“Predictor reduction was based on a B-H FDR threshold of 5% of Pearson correlation using a stepwise general linear model accounting for age, sex and white blood counts and including all 72 miRNAs associated with parasitemia. Model selection was based on the Akaike information criterion. A set of four miRNAs were identified as the best predictive variables (hsa-miR-199-3p, hsa-miR-3173-5p, hsa-

miR-342-3p and hsa-miR-532-5p). Two-fold cross-validation was done by randomly shuffling the dataset into training and test sets (60 and 40% of the full dataset, respectively). Three metrics were used to assess the performance of the model for both training and test sets. First, the Root Mean Squared Error (RMSE) of the model. A lower RMSE value indicates a better fit. Second, Harrell's C-Statistic was used to estimate the probability of concordance between predicted and observed parasitemia values. Harrell's C-Statistic values range between 0.5 and 1 and a value of 0.5 indicates a total disagreement between predicted and observed parasitemia values. Third, Pearson correlation between predicted and observed parasitemia values. High Pearson correlation values indicate a good performance of the model."

Results:

"In a stepwise general linear model accounting for sex, age and white blood cell count and using a 2-fold cross validation by randomly shuffling the dataset into training and test sets (60 and 40% of the full dataset, respectively), four miRNAs (hsa-miR-199-3p, hsa-miR-3173-5p, hsa-miR-342-3p and hsa-miR-532-5p) significantly predicted log₂ parasitemia in the test dataset (B-H FDR<0.05, Pearson correlation = 0.73, RMSE = 2.1983, Harrell's C-Statistic = 0.8238) (see Methods and Supplementary Figure S3)."

2. The nominally significant (permutation corrected) p-value from the miR-eQTL analysis should be moved to the supplements and the results section should focus on the FDR-corrected findings.

Reply: As requested, permutation corrected p-values are moved to the supplements. Table 1 and the results section now focus on FDR-corrected findings.